# Structure and mechanism of a phosphotransferase system glucose transporter

Patrick Roth [1], Jean-Marc Jeckelmann[1], Inken Fender[1], Zöhre Ucurum[1], Thomas Lemmin [1] & Dimitrios Fotiadis [1] ✉

Glucose is the primary source of energy for many organisms and is efficiently taken up by bacteria through a dedicated transport system that exhibits high specificity. In *Escherichia coli*, the glucose-specific transporter IICB$^{Glc}$ serves as the major glucose transporter and functions as a component of the phosphoenolpyruvate-dependent phosphotransferase system. Here, we report cryo-electron microscopy (cryo-EM) structures of the glucose-bound IICB$^{Glc}$ protein. The dimeric transporter embedded in lipid nanodiscs was captured in the occluded, inward- and occluded, outward-facing conformations. Together with biochemical and biophysical analyses, and molecular dynamics (MD) simulations, we provide insights into the molecular basis and dynamics for substrate recognition and binding, including the gates regulating the binding sites and their accessibility. By combination of these findings, we present a mechanism for glucose transport across the plasma membrane. Overall, this work provides molecular insights into the structure, dynamics, and mechanism of the IICB$^{Glc}$ transporter in a native-like lipid environment.

Carbohydrates play a vital role in bacterial physiology, providing energy for cellular life and serving as important carbon source for the synthesis of a wide variety of biomolecules. To increase the competitiveness and ensure survival in rapidly changing environments with different carbohydrate supplies, bacteria developed a variety of dedicated transport systems for their uptake[1]. D-glucose (Glc; hereinafter referred to as glucose), the preferred carbon source for many bacterial species, is primarily acquired by the glucose-specific phosphoenolpyruvate-dependent phosphotransferase system (PTS). The PTS is a unique multi-component group translocation system that couples vectorial transmembrane transport with concomitant phosphorylation of carbohydrates[2-4]. Its ubiquitous distribution in bacteria and complete absence from eukaryotes renders it an attractive target for antimicrobial agents. Besides its primary metabolic role in carbohydrate transport, phosphorylation, and chemoreception[5,6], the PTS is involved in diverse functions such as gene[7] and metabolic regulation[8], biofilm formation[9], antibiotic resistance, and virulence[10-12]. Unlike other systems, energy originates from the glycolytic intermediate phosphoenolpyruvate. Its phosphoryl group is extracted and transferred by cytoplasmic soluble system I components, i.e., the general energy-coupling enzyme I (EI) and the histidine-containing phosphocarrier protein, and ultimately brought to the substrate-specific system II (EII). EII is a multi-modular protein complex, typically composed of three (partially interconnected) units: IIA, IIB, and IIC. These units can exist as separate polypeptides, as seen in the *N,N'*-diacetylchitobiose-specific PTS EII from *E. coli*. Alternatively, some units can be fused into a single polypeptide chain, like the glucose-specific PTS system in the same organism. In the latter case, the system consists of a soluble protein, IIA$^{Glc}$, and a two-domain membrane transporter IICB$^{Glc}$. The IIC$^{Glc}$-domain is the membrane-embedded glucose transporter domain, and IIB$^{Glc}$ is the soluble cytoplasmic protein domain responsible for phosphorylating glucose[13]. Upon vectorial IICB$^{Glc}$-mediated substrate transport, glucose is covalently modified, resulting in the formation of intracellular glucose-6-phosphate that may then serve as a direct substrate for the glycolytic pathway[14,15] (Fig. 1a). The transport activity is ultimately coupled to transcriptional regulation by the

[1]Institute of Biochemistry and Molecular Medicine, Medical Faculty, University of Bern, Bern, Switzerland. ✉e-mail: dimitrios.fotiadis@unibe.ch

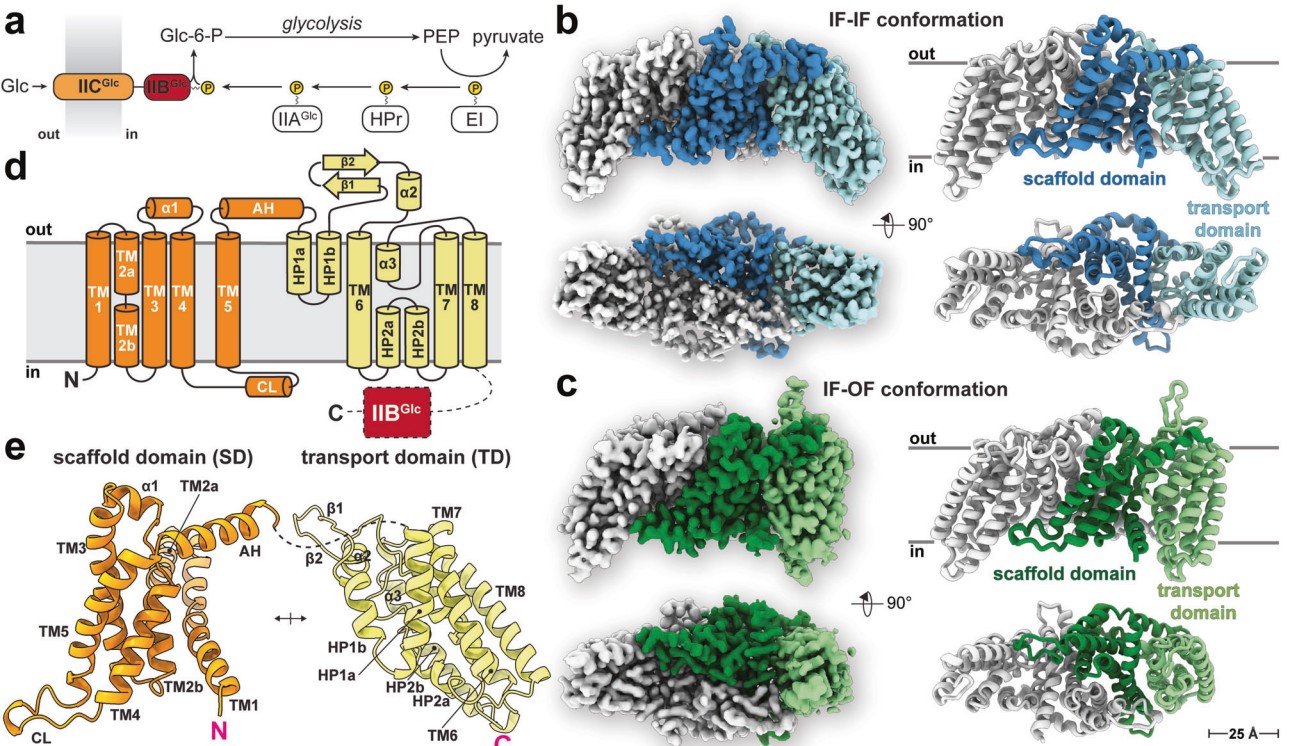

**Fig. 1 | Function, topology, and overall structures of IICB^Glc conformations.**
**a** Schematic overview of the glucose-specific phosphoenolpyruvate-dependent phosphotransferase system (PTS). The soluble enzyme cascade (EI, HPr, and IIA^Glc) extracts and shuttles the phosphoryl group from phosphoenolpyruvate (PEP) to the membrane-bound permease IICB^Glc, which transports and concomitantly phosphorylates the incoming glucose (Glc). Cryo-EM density map of (**b**) the symmetric IICB^Glc inward-facing dimer (IF-IF) and (**c**) the asymmetric inward- and outward-facing dimer (IF-OF), contoured at a threshold level of 0.04. In one protomer, the scaffold and transport domains are colored with different shades of blue

and green, respectively. The transporter is shown from the membrane plane (*top*) and cytosolic face (*bottom*), and the corresponding models (*right*) are shown in cartoon representation. The lipid bilayer interface is indicated with horizontal lines. **d** Schematic topology diagram of IICB^Glc with the scaffold domain (SD) colored in orange and the transport domain (TD) in yellow. The unresolved linker and IIB^Glc protein domain (red) are shown as dashed lines. **e** Cartoon representation of the SD and TD structures labeled as in (**d**). The protomer is broken open at the SD/TD domain interface for better visualization.

repressor Mlc via a direct interaction[16], highlighting the role of IICB^Glc in chemoreception. The glucose-specific PTS IICB^Glc protein is a member of the glucose-glucoside transporter family (transport classification accession 4.A.1)[17] and the most prominent within the GFL (glucose/fructose/lactose) superfamily. Crystal structures of GFL-superfamily members, i.e., the maltose- and *N,N'*-diacetylchitobiose-specific transporters, have been reported previously[18–20]. Despite numerous attempts using X-ray and electron crystallography[21–24], no structure for the relevant glucose-specific transporter is currently available, despite most of the functional understanding of the PTS being derived from studies on the glucose-specific PTS.

In this study, we use cryo-electron microscopy (cryo-EM) to reveal the architecture and structure of the glucose-specific PTS transporter IICB^Glc from *E. coli*. We reconstitute IICB^Glc in lipid nanodiscs and obtain two high-resolution structures of the substrate-bound inward- and outward-facing occluded conformations of the transporter within a native-like lipid environment. Supported by molecular dynamics simulations, we provide insights into the transport mechanism and the molecular basis for glucose recognition, binding, and transport. In summary, these findings establish a framework for addressing long-standing questions on the mechanistic basis of the primary glucose transporter in bacteria.

## Results and discussion
### Structure determination of the IICB^Glc protein
To elucidate the structural basis of carbohydrate transport by the glucose-specific PTS transporter, we conducted cryo-EM experiments

using purified IICB^Glc protein from *E. coli* in the presence of glucose. A native-like lipid environment was established through lipid nanodisc reconstitution of pure, monodisperse full-length protein (Supplementary Fig. 1a–d). After image acquisition and processing, single-particle analysis using cryoSPARC was performed revealing particles with secondary structure features in two-dimensional (2D) class averages. Further classification converged into distinct populations of particles (Supplementary Fig. 1e–g), which were refined to two three-dimensional (3D) density maps with global resolutions of 2.6 and 2.9 Å according to the gold-standard Fourier shell correlation criterion (Supplementary Fig. 2). Inspection revealed that these structures correspond to the membrane-embedded, dimeric IIC^Glc transport protein domain. The absence of a defined density of IIB^Glc indicates intrinsic flexibility of this domain. Hence, atomic models of glucose-bound IIC^Glc were built into the 3D density maps revealing two distinct conformational states (Supplementary Table 1).

### Overall structure of the dimeric IIC^Glc transporter
The IIC^Glc protein domain mediates substrate translocation and is almost completely embedded in the lipid bilayer. IIC^Glc forms an ellipsoidal homodimer assembled around a central axis (Fig. 1b, c). Individual protomers are composed of eight transmembrane segments, one amphipathic helix (AH), two α-helical hairpins (HP1 and HP2) and three prominent loops: Two periplasmic and one cytoplasmic (schematically shown in Fig. 1d). Protomers can be divided into an N-terminal (TM1-5) and a C-terminal (TM6-8) functional domain, representing the central scaffold domains (SD) and the peripheral

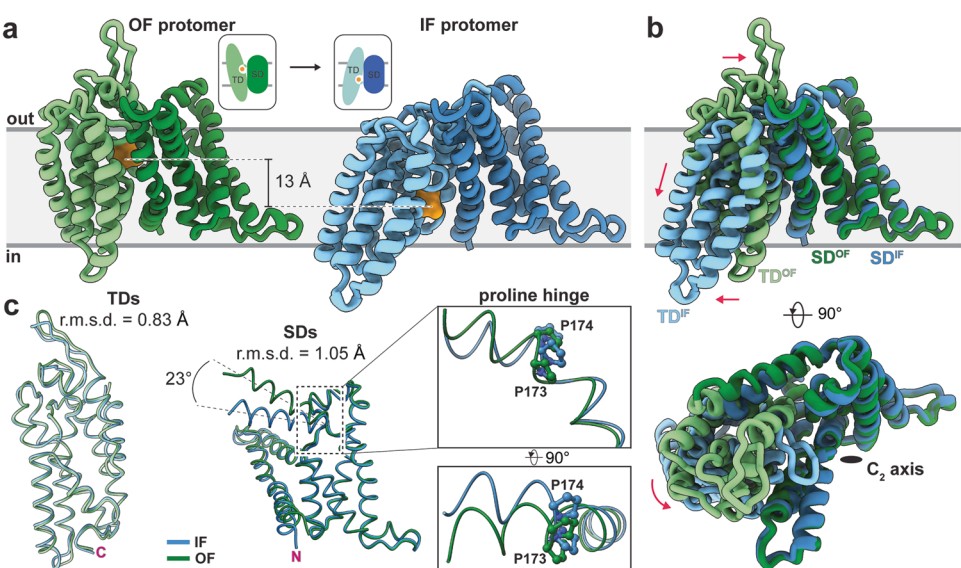

**Fig. 2 | The elevator-type transport mechanism of IIC^Glc. a** Visualization of the elevator movement, i.e., the rigid-body translocation of the transport domain (TD) relative to the scaffold domain (SD), by a side-to-side comparison of both extreme states. The location of the glucose binding site is indicated (orange), and the position of the ring oxygen was used to measure the indicated vertical translocation distance. The inset shows a schematic representation of the conformational change. **b** View from the membrane plane (*top*) and periplasm (*bottom*) of the superimposition of the outward-facing (OF; SD in dark green and TD in light green) and inward-facing (IF) state protomers (SD in dark blue and TD in light blue) aligned on the quasi-static SD. Red arrows indicate the movements of the TD upon conformational change. **c** Structural alignment of SDs (residues 4-173) and TDs (residues 191-386) in IF and OF conformations: Calculated root-mean-square deviations (r.m.s.d.) are indicated. Note the substantial change in angle originating from the proline hinge between TM5 and AH.

transport domains (TD) in the dimer, respectively. The SDs form the dimerization interface, whereas the TDs are involved in substrate binding and translocation. The AH, a C-terminal, horizontal extension of TM5, connects the SD and TD. HP1 is located after AH, and HP2 is situated between TM6 and TM7. The first prominent periplasmic loop consists of a short α-helix (α1), while the second is comprised of a β-hairpin followed by a short α-helix (α2) (Fig. 1d, e). The cytoplasmic loop (CL) is located between TM4 and TM5, and adopts an α-helix-like structure extending towards the TD of the adjacent protomer. The third short α-helix (α3) lies between TM7 and TM8, and extends HP2a vertically (Fig. 1d, e). The protomers are intertwined in a handshake-like manner and share an extensive interface of ~2800 Å² predominantly formed by the N-terminal SDs of each protomer, involving interactions between TM1, TM2, TM3, TM5, and CL. Specifically, the dimerization interface is characterized by prevailing hydrophobic interactions, and further polar contacts such as a salt bridge (E50'/K91) and hydrogen bonds (N74'/D75 and N74'/G76(N), and S21'/R151) (prime indicates opposite protomer, Supplementary Fig. 3a). Interestingly, the predicted AlphaFold2 IIC^Glc oligomer was consistent with the experimentally observed dimeric assembly (Supplementary Fig. 3b, c). Amino acid sequence analysis revealed that IIC^Glc and selected IIC proteins, which are specific for other carbohydrates, share moderate sequence identities (Supplementary Fig. 4a). IIC^Glc shares a similar structural arrangement as the previously reported structures of maltose-(IIC^Mal)[18,19] and *N,N'*-diacetylchitobiose-specific IIC transporter (IIC^Chb)[20]. While the protomers exhibit a comparable topology, subtle differences are discernible in the periplasmic regions (Supplementary Fig. 4b–d).

**Major conformational rearrangements imply an elevator-type transport mechanism**

Utilizing nanodisc reconstitution in lipid bilayers, we successfully captured the IIC^Glc transporter in two functionally relevant conformational states: (i) a symmetric dimer structure with both protomers in the inward-facing state (IF-IF), and (ii) an asymmetric dimer structure with one protomer in the inward-facing and the other in the outward-facing state (IF-OF) (Fig. 1b, c). This enabled us to characterize the major conformational changes between outward- and inward-facing states (Fig. 2a, b). Superposition of IF- and OF-TDs as well as IF- and OF-SDs revealed only minor structural changes reflected in relatively low root-mean-square deviation (r.m.s.d.) values of 0.83 Å for TDs and 1.05 Å for SDs (Fig. 2c, Supplementary Fig. 5a). IF protomers of both, symmetric and asymmetric structures are comparable (r.m.s.d. of 0.59 Å). The global structural transition is characterized by a substantial rigid body motion of the TD, resulting in a vertical shift of ~13 Å for the substrate binding site. This alternating mechanism presents the substrate binding site to either the cytoplasmic or the periplasmic side of the membrane (Supplementary Video 1). In summary, the structural elements and the observed conformational states exhibit typical hallmarks of an elevator-type transporter[25], suggesting that IIC^Glc utilizes this mechanism for substrate translocation. The TD and SD domains are connected by TM5-AH (Fig. 1d), which contains the P173/P174 double proline motif that acts as a hinge point, as evident from the comparison of the IF and OF states (Fig. 2c). Due to the movement of the TD, AH is rearranged and the angle between TM5 and AH increases by 23°. The conformational change alters the interaction interfaces between SD and TD. Although the main interaction contribution remains hydrophobic, e.g., originating from aliphatic residues, several hydrogen bond pairs, i.e., S34/N350, G53(O)/Q231 and Y188/E224(N) in the IF state, and F146'(N)/S286 and V118(N)/G209(O) in the OF state (O and N in parentheses denote the backbone carbonyl oxygen and nitrogen atoms, and prime indicates opposite protomer), provide stabilization of the respective states (Supplementary Fig. 5b, c). In addition to these polar interactions, the aromatic ring of F337 is engaged in a π-π stacking interaction with either F56 in the OF conformation or F146 in the IF conformation. F337 hydrophobically shields the substrate binding site and potentially acts as a thin gate[26], preventing substrate diffusion by occlusion (Supplementary Fig. 5d). From an electrostatic perspective, the SD/TD interface is differentially charged on the cyto- and periplasmic side (Supplementary Fig. 5e).

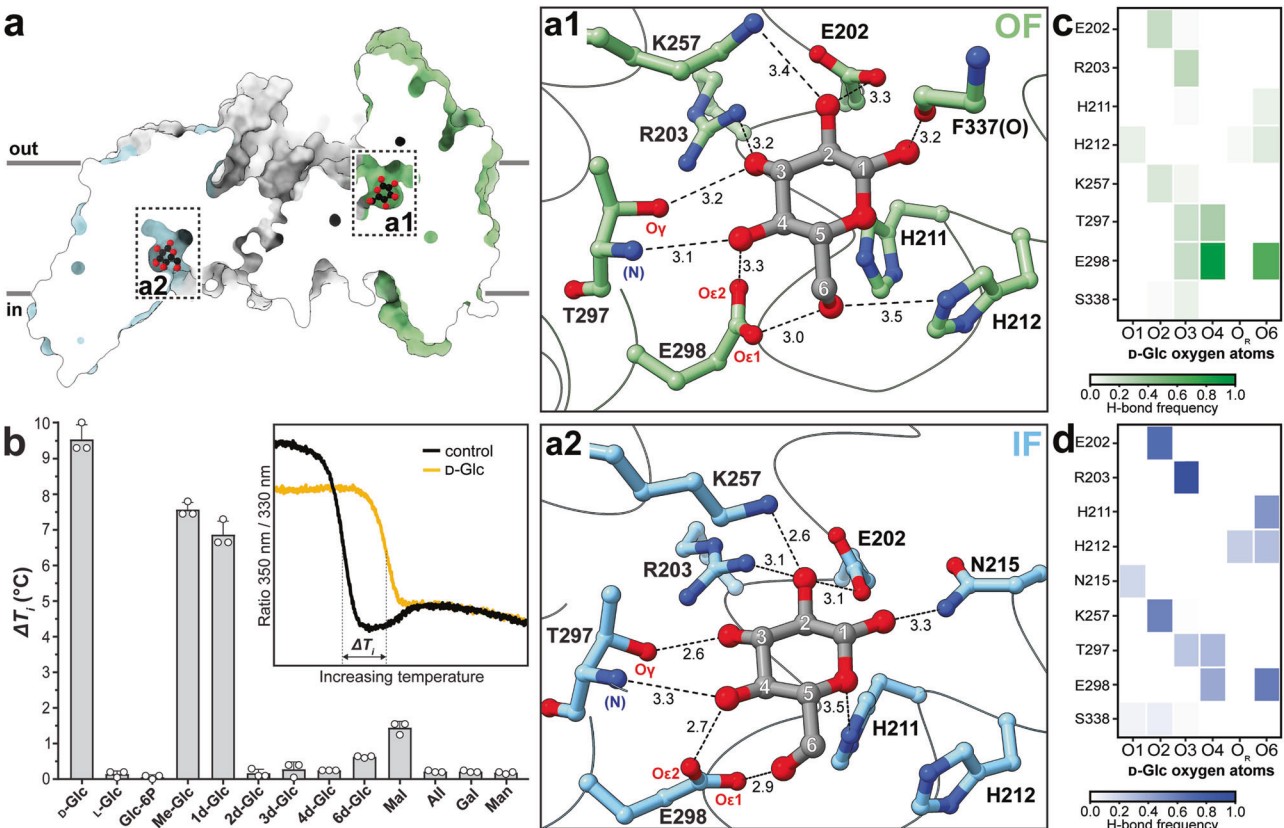

**Fig. 3 | Characterization of the glucose binding sites and ligand-binding mechanism in IIC^Glc. a** Sagittal cross-section of the asymmetric inward- and outward-facing (IF-OF) structure, with the glucose binding sites highlighted. Enlarged view on the substrate binding sites displaying interactions (≤3.5 Å) with glucose in the OF (**a1**) and IF states (**a2**). Hydrogen bond distances are indicated in Å, O, and N in parentheses denote the backbone carbonyl oxygen and nitrogen atoms, respectively. **b** Thermal stabilization of IICB^Glc upon binding of different ligands ($\Delta T_i$), measured by nano-differential scanning fluorimetry (nanoDSF; the inset shows representative melting curves for the control and in presence of 10 mM

D-glucose (D-Glc)). Data are shown as the mean ± standard deviation; $n = 3$ biological replicates, and abbreviations for ligands are detailed in the Supplementary Fig. 6. **c, d** Frequency of hydrogen bonds (H-bond frequency) formed between the oxygen atoms of the bound glucose molecule (O1–O4 and O6 represent the five D-Glc hydroxyl groups, and $O_R$ the ring oxygen) and amino acids of the OF and IF substrate binding site in IICB^Glc, respectively. H-bond frequencies were averaged across all five replicates of the MD simulations, and color coded according to their frequency.

## Structural basis for ligand recognition, binding, and specificity

The initial step in the transport cycle is the specific recognition and binding of the glucose substrate by the IIC^Glc protein domain. Each IIC^Glc protomer contains a substrate binding site situated in the TD, positioned proximally to the interface between the TD and SD. It forms a near-spherical cavity composed of residues from HP1a and HP1b, HP2b, TM6, and the TM7-α3 loop to accommodate the substrate. In the IF conformation, the position of the substrate binding site relative to the membrane plane faces the cytosolic side, whereas in the OF conformation, it faces the periplasmic side (Fig. 3a). Additional Coulomb potential density within the polar substrate binding site was unambiguously assigned to the substrate glucose, which is coordinated by multiple proximal polar residues. The overall architecture of the glucose binding site is comparable in both conformations, with only a few minor differences in the molecular coordination. In both cases, the glucose hydroxyl group 6 (OH-6) is oriented toward the cytoplasm and is coordinated by Oε1 of E298 (Fig. 3a1, a2) and additionally weakly by H212 in the OF conformation. In both conformations, OH-2, OH-3, and OH-4 form each multiple hydrogen bonds with neighboring amino acids, including E202, R203, K257, T297, and E298 (Fig. 3a1, a2). A major difference between the IF and OF states is observed for the glucose OH-1 binding, which is coordinated by the F337 carbonyl oxygen in OF and by the N215 side chain in IF. The ring oxygen atom is weakly coordinated by H211 in the IF state. The hydrogen atoms attached to glucose C-5 and C-6 carbon atoms face a hydrophobic

patch in the substrate binding site provided by F337. To corroborate the molecular contribution of individual hydroxyl groups on substrate coordination, we performed a thermostability assay using selected ligands, i.e., all possible deoxy-D-glucose analogs, and selected glucose epimers and isomers. This innovative assay provides significant advantages over the conventional mutagenesis approach, due to its speed and the elimination of risks associated with disrupting the substrate binding site or impairing expression of mutant IICB^Glc versions. The thermal stability of IICB^Glc in the presence of potential ligands was measured by label-free nano-differential scanning fluorimetry (nanoDSF), which yields an increased inflection temperature ($T_i$) in the case of a stabilizing interaction compared to the no-ligand reference (i.e., a positive $\Delta T_i$). As expected, D-glucose but not L-glucose induced a significant thermal stabilization effect of almost 10 °C (Fig. 3b), while the PTS-product glucose-6-phosphate (Glc-6P) had no thermostabilizing effect. Among the deoxy-D-glucose analogs tested, 1-deoxy-glucose and 1-O-methyl-glucose bound best, suggesting that IICB^Glc tolerates some variability at the OH-1 position. Conversely, 2-, 3-, 4- and 6-deoxy-glucose showed only residual stabilization, indicating that these hydroxyl groups are essential for glucose binding and therefore act as specificity determinants. Neglectable stabilization was observed with other carbohydrates tested, i.e., allose (All), galactose (Gal) and mannose (Man) (Fig. 3b). In contrast, D-maltose, a disaccharide composed of two α(1 → 4)-linked glucose molecules, induced a slight stabilization of IICB^Glc. The ligand

binding sites of IIC[Mal] [19] and IIC[Glc] (Supplementary Fig. 4e, f) share some similarities: The glucopyranosyl moiety of maltose is comparably coordinated, interacting with E355, T354, R232, E231, and H240. In contrast, glucose is coordinated in IIC[Glc] by additional interactions, e.g., with N215. *N,N'*-diacetylchitobiose coordination by IIC[Chb] [20] is significantly different, with only E334 and H250 being conserved (Supplementary Fig. 4g).

## Molecular dynamics simulations provide insights into the dynamics of substrate coordination

To deepen our understanding of substrate coordination and binding to IIC[Glc], we conducted five independent replicates of molecular dynamics (MD) simulations, seeded from the glucose-bound IF-OF model embedded in a lipid bilayer, for a cumulative simulation time of 4.85 μs (Supplementary Fig. 7). Interestingly, the glucose molecule exhibited varied residence times within the different binding pockets. In all replicates, it displayed a transient association with the OF substrate binding site, remaining bound for only ~23 ± 10 ns before exiting. In contrast, glucose exhibited a notably extended residence time within the IF substrate binding pocket, remaining bound throughout the simulation in two replicates, while dissociating after ~152 ± 90 ns in the other three simulations. To further characterize the interactions between the glucose and the coordinating amino acids, we calculated the hydrogen bond frequencies between each glucose oxygen atom and surrounding residues within the binding site. Our analyses revealed a strong correlation with the structural and biophysical data. Within the OF substrate binding pocket, the glucose molecule engages extensive interactions with all the residues identified in the experimental cryo-EM structure. Notably, the conserved E298 and T297 emerged as the primary partners for hydrogen bond interactions with the glucose molecule, frequently interacting with OH-3, OH-4, and OH-6. The glucose coordination is slightly rearranged after the elevator movement of the TD in the IF substrate binding site. Specifically, interactions of amino acid residues with OH-2 and OH-3 involving E202, R203, and K257, and interactions between OH-6 and H211, H212 and again E298 arise. Collectively, these findings reinforce previous observations and provide valuable insights into the dynamic changes in substrate coordination between the IF and OF states of the IIC[Glc] protein.

## HP1b movement and a full glucose binding event

To enhance our understanding of the transport mechanism at the molecular level, investigation of the initial step of substrate recognition and binding to the transporter is essential. Therefore, we extended our MD simulation-based analysis of the OF state and found a correlative motion pattern upon release of the glucose molecule across all five replicas: HP1b, which delimits the lateral periplasmic access to the binding site and contains residues involved in substrate coordination, undergoes a directed tilting motion away from its original position in the cryo-EM structure towards the AH (Fig. 4a). The maximal observed angle changes prior to glucose release range approximately from 10° to 15° (Fig. 4b), along with a center of mass shift of 2.9 ± 0.8 Å. In addition, in three of the five replicates, helix α2 transitions from a previously nearly horizontal orientation to a vertically aligned configuration by the end of the simulations, corresponding to a center of mass shift of 5.4 ± 2.8 Å. Following glucose release, a further shift of HP1b was observed, converting the transporter to a putative outward open state.

To further investigate glucose binding, we identified 15 pre-binding events from the MD simulations, defined by a glucose molecule within 3 Å of residues H211 and H212, and employed these structures as seeds for simulations with shorter time steps. In one simulation, a full-binding event was observed, during which a glucose molecule entered the OF substrate binding pocket and remained bound for ~130 ns (Supplementary Video 2). While the binding

orientation closely resembled that found in the cryo-EM structure (r.m.s.d. of 2.3 Å), the network of interacting hydrogen bonds exhibited subtle variations (Fig. 4c, d). Specifically, the glucose molecule forms hydrogen bonds with the amino acids R203, K257, G295, T297, E298, and S338. E298 and T297 remained the predominant hydrogen bond interaction partners with the glucose molecule, similar to the cryo-EM structure. Differences between simulated and experimental position of the glucose molecule are due to a slight rotation of the glucose, and the displaced HP1b away from the substrate binding site. This results in the stabilization of the glucose OH-2 by G295(O) and S338 (Fig. 4c, d) instead of E202 and K257 (Fig. 3c). These subtle variations in interactions suggest that the MD simulation might have captured a conformation representing a substrate-bound outward open state, providing insights into the early stages of the binding process. It is worth noting that enhanced sampling MD simulations could be employed in future studies to further explore the conformational landscape and free energy profile of glucose binding. Consistent with the physiological role of IIC[Glc], the glucose in the IF state, awaiting phosphorylation by IIB[Glc] displayed enhanced stability, remaining bound throughout the simulation in two replicates. In the other replicates, the release of glucose was coupled with a general opening of the substrate binding site, which involved the CL of the opposite protomer.

## Gates in IIC[Glc] modulate substrate accessibility

Taken together, these observations raise questions about the precise conformational states captured in experimental and simulated structures. We used the program HOLE[27] to quantify the pore radii along the exit pathways, revealing that the substrate binding sites in the cryo-EM structures are minimally accessible to the corresponding surrounding environment (Fig. 4e, h). Considering the size of glucose (~4.7 Å for O-1⋯O-3 distance) and the narrowest pore diameters of ~3.6 Å, the glucose-bound states obtained by cryo-EM likely represent the substrate-bound outward-facing, occluded (OF[occluded-holo]) and substrate-bound inward-facing, occluded (IF[occluded-holo]) states. Conversely, the substrate unbound states of the OF protomer from the MD simulations displayed progressively widening average pore radius toward the bulk, representing a potential substrate-free outward open state (OO[apo]; Fig. 4f). Similarly, simulations with glucose release in the IF protomer showed increasing diameters, potentially representing a substrate-free inward open state (IO[apo]), whereas simulations with persistent glucose binding in the IF pocket resemble the cryo-EM structure in the IF[occluded-holo] state (Fig. 4i). Key gating residues, responsible for pore constriction, were identified in both states. In the OF[occluded-holo] state, these include H212, Q219 (both part of HP1b), and F337 (Fig. 4g and Supplementary Fig. 5d). In the IF[occluded-holo] state, these involve M17, H211, H212, E298, and crucially, F146' from the CL of the opposite protomer (Fig. 4j).

This study provides structural insights into the substrate recognition, binding, and transport mechanism of the glucose-specific PTS transporter IICB[Glc], which plays a pivotal role in bacterial carbon and energy acquisition. By presenting cryo-EM structures of IICB[Glc] reconstituted in a lipid bilayer, we revealed the IIC[Glc] transmembrane transporter's distinct IF and OF conformations, both bound to glucose in an occluded state, as well as the dynamic flexibility of the cytosolic IIB[Glc] domain. Significantly, the high-resolution cryo-EM structures provided a detailed view of the elevator-type transport mechanism advancing our molecular understanding. Further, the observed protomer asymmetry in the IF-OF IIC[Glc] transporter dimer structure suggests that each protomer functions independently. MD simulations using the obtained cryo-EM structures have broadened our view of the conformational landscape, unveiling additional potential states critical for the transporter's function such as the outward and inward open states. The comprehensive model of the IICB[Glc] transport cycle, as illustrated in Fig. 5, summarizes the current structural and mechanistic

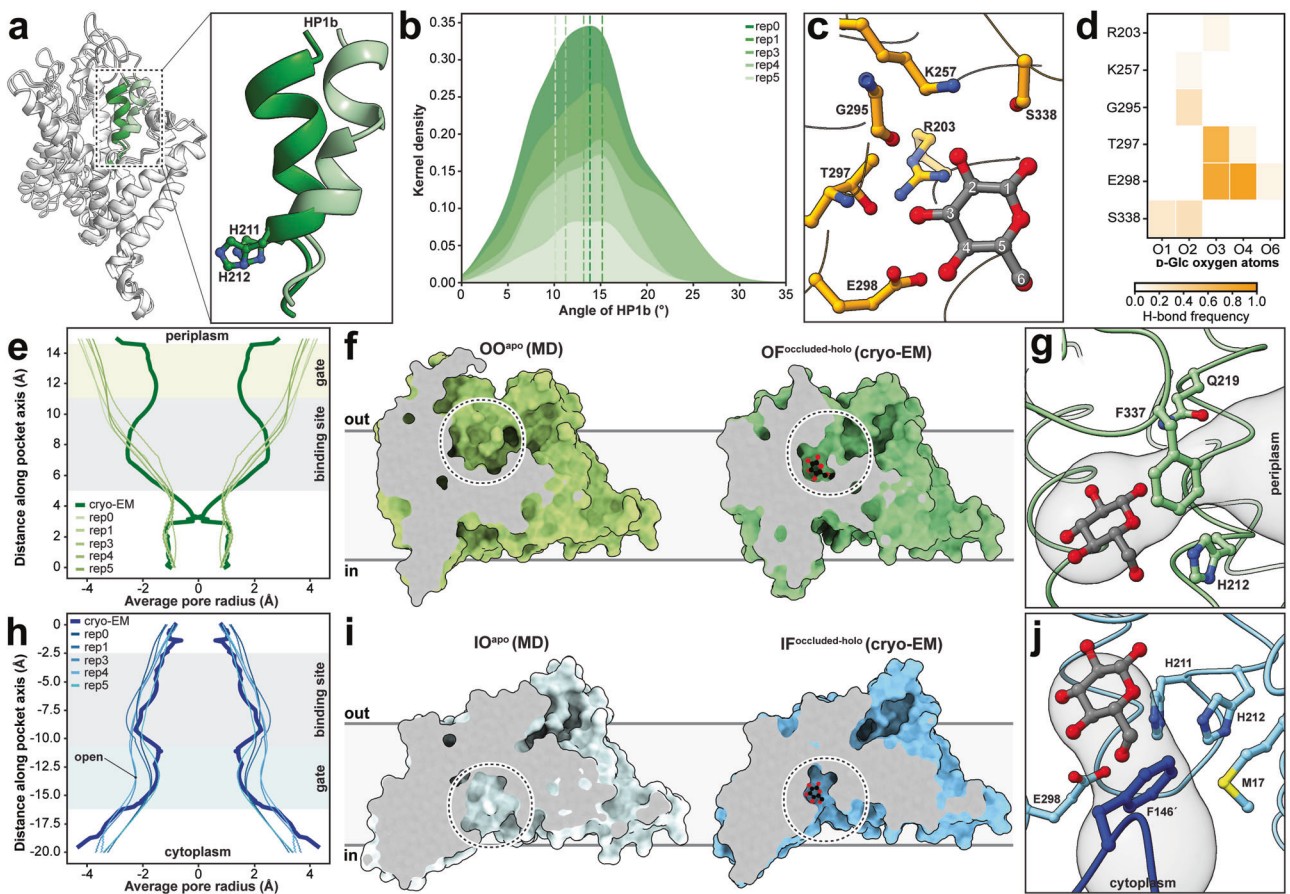

**Fig. 4 | Dynamics of IIC^Glc determined by MD simulations. a** Visualization of a representative HP1b displacement in the OF state observed during MD simulations. The cryo-EM and MD simulation structures are shown in dark and pale green, respectively. **b** Kernel density estimation (KDE) plot depicts the distribution of changes in the HP1b angle compared to the cryo-EM structure. The dashed lines highlight the maximum angle deviations observed before glucose release in the five replicates. **c** Representative frame of the observed full-binding event of glucose to the OF state with highlighted frequent amino acid interaction partners. **d** Hydrogen-bond frequency plot of the full-binding event showing the emerging binding amino acids during the bound state (~130 ns). **e, h** Average pore radii for the

experimental cryo-EM and MD simulated OF and IF states, respectively. The location of the binding site and of the gating residues is indicated. **f, i** Sagittal views on the substrate binding site in the four different observed states: Substrate-free outward open (OO^apo; from MD simulations), substrate-bound outward-facing, occluded (OF^occluded-holo; from cryo-EM), substrate-free inward open (IO^apo; from MD simulations) and substrate-bound inward-facing, occluded (IF^occluded-holo; from cryo-EM). **g, j** Gating residues that restrict the accessibility of the substrate binding sites towards periplasm (OF^occluded-holo structure) and cytosol (IF^occluded-holo structure), respectively.

comprehension of this system. In summary, our findings provide molecular insights into the structure and function of IICB^Glc, contributing to the understanding of its transport mechanism.

## Methods

### IICB^Glc overexpression and membrane isolation
The gene encoding the wild-type glucose-specific transporter IICB^Glc (UniProt ID: P69786) from *E. coli* present in the pTSG plasmid[24] was re-cloned and inserted into the pZUDF plasmid[28]. This resulted in a construct encoding full-length IICB^Glc with a decahistidine tag, which was transformed into the *ΔarcBΔptsG*-BW25113 *E. coli* strain (made in-house using the parent *ΔarcB*-BW25113 strain[29]). Transformed bacteria were selected in 100 μg/mL ampicillin-containing LB medium (Luria Bertani), aliquoted, and stored in 20% (w/v) glycerol at −80 °C. For large-scale IICB^Glc overexpression, growth media were inoculated with an overnight preculture at a 1:100 ratio and cultured at 250 rpm at 37 °C in an incubator shaker (Multitron, Infors HT) until an optical density at 600 nm (OD_600) of about 1 was reached. Overexpression was induced by addition of 0.2 mM isopropyl-β-D-1-thiogalactopyranoside (IPTG; final concentration) to the cell cultures and subsequent growth for 4 h. Cells were collected by centrifugation at 10,000 × *g* for 10 min and the resulting cell pellets resuspended in lysis buffer (50 mM Tris-HCl pH

8.0, 300 mM NaCl) and centrifuged again before storage at −80 °C. For membrane isolation, cells were thawed and resuspended in lysis buffer supplemented with 10 mM EDTA. Cells were lysed by shear forces using a Microfluidizer (MP-110, Microfluidics) by passing them through five rounds at 1500 bar. Cell debris and unlysed cells were removed by centrifugation (10,000 × *g*, 10 min, 4 °C) and membranes residing in the supernatant were pelleted by ultracentrifugation (150,000 × *g*, 60 min, 4 °C). Membranes were washed by homogenization in ice-cold membrane buffer (40 mM HEPES [4-(2-hydroxyethyl)-1-piperazineethanesulfonic acid]-NaOH pH 8.0, 300 mM NaCl) using a glass stir homogenizer and again subjected to ultracentrifugation (150,000 × *g*, 60 min, 4 °C) before final resuspension in membrane buffer at 200 mg/mL. The suspension was aliquoted and flash-frozen in liquid nitrogen for storage at −80 °C until further use.

### Purification of IICB^Glc
All subsequent purification steps were conducted at 4 °C. For a typical purification, 2 mL of membranes corresponding to ~0.5 L of cell culture were solubilized for 1.5 h on a rotating wheel in 7 mL membrane buffer supplemented with 5 mM β-mercaptoethanol (β-ME), 2% (w/v) DDM (*n*-dodecyl-β-D-maltopyranoside, Glycon) and 5 mM glucose. Detergent-insoluble material was removed by ultracentrifugation (100,000 × *g*,

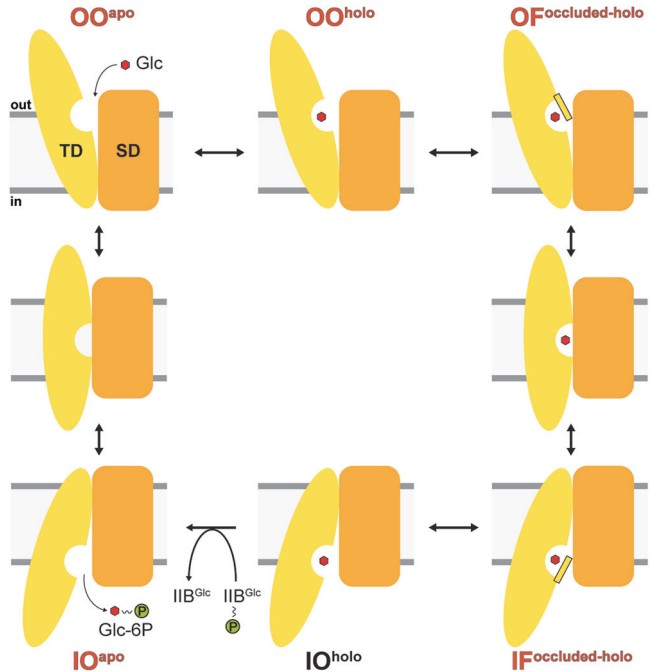

**Fig. 5 | Schematic illustration of the proposed glucose transport mechanism by monomeric IIC$^{Glc}$,** focusing on the major steps of the transport cycle, i.e., glucose (red hexagon) binding, gating, and the rearrangements of the transport domain (TD, yellow) relative to the scaffold domain (SD, orange). The thin gates are depicted in yellow and outlined in black. Abbreviations used: Substrate-free outward open (OO$^{apo}$), substrate-bound outward open (OO$^{holo}$), substrate-bound outward-facing, occluded (OF$^{occluded-holo}$), substrate-bound inward-facing, occluded (IF$^{occluded-holo}$), substrate-bound inward open (IO$^{holo}$), substrate-free inward open (IO$^{apo}$), D-glucose (Glc) and D-glucose-6-phosphate (Glc-6P). The green disc contoured in black with a "P" symbolizes a phosphoryl group. Experimental states from cryo-EM and simulated states from MD simulations determined in this work are highlighted in red text color.

30 min, 4 °C). The supernatant was diluted 1:1 with wash buffer (40 mM HEPES-NaOH pH 8.0, 300 mM NaCl, 5 mM β-ME, 20 mM imidazole, 0.02% (w/v) DDM, 5 mM glucose) and supplemented with 500 µL equilibrated Ni$^{2+}$-NTA resin (Qiagen) for batch binding. This suspension was transferred to an Econo column (Bio-Rad), and the resin was washed with 60 column volumes of wash buffer. The protein was eluted with elution buffer (wash buffer with 300 mM imidazole). Protein-containing fractions were pooled and concentrated in an Amicon Ultra-4 50 kDa spin column concentrator (Millipore). The concentrated protein was centrifuged (20,000 × g, 10 min, 4 °C) and applied to a Superdex 200 increase 10/300 GL size-exclusion chromatography column equilibrated with 20 mM HEPES-NaOH pH 8.0, 150 mM NaCl, 5 mM β-ME, 0.02% (w/v) DDM, 5 mM glucose connected to an ÄktaPure chromatography system (Cytiva). Peak fractions containing IICB$^{Glc}$ were pooled, concentrated using an Amicon Ultra-4 50 kDa concentrator and immediately used for downstream applications.

### Differential scanning fluorimetry

For thermostability analysis of the IICB$^{Glc}$ protein, purification was essentially performed as described above, except that no glucose was added to the buffers. Test ligands including glucose (Supplementary Fig. 6b) were dissolved in water and then added to the protein in 20 mM HEPES-NaOH pH 8.0, 150 mM NaCl, 5 mM β-ME, 0.02% (w/v) DDM, resulting in a final protein concentration of 4 µM and 10 mM of the respective ligand. After incubation for 5 min, the thermostability was measured using the Tycho NT.6 apparatus (NanoTemper, Germany). ~8 µL of each sample was aspirated into a glass capillary,

which was then heated in a range from 35 °C to 95 °C (30 °C/min heating rate). The change in the ratio of the intrinsic fluorescence intensities at 350 nm and 330 nm was used to determine the thermal unfolding transitions. Inflection temperature difference values (Δ$T_i$) were obtained by subtracting the average inflection temperature of the control (protein plus H$_2$O) from the average inflection temperatures obtained from samples with the corresponding carbohydrates (Fig. 3b). Experiments were repeated three times (n = 3) from different purifications, and each measurement was performed as a technical triplicate. Data were processed in GraphPad Prism version 9.0 (GraphPad) and are presented as the mean ± standard deviation.

### Overexpression and purification of MSP1E3D1

The membrane scaffold protein MSP1E3D1 (MSP) was overexpressed and purified, essentially as described previously[30]. The pET28a plasmid (Addgene plasmid #20066) encoding N-terminal hexahistidine-tagged MSP was transformed into *E. coli* BL21(DE3) cells. Cells were cultivated in terrific broth medium supplemented with 50 µg/mL kanamycin at 37 °C and 180 rpm (Multitron, Infors HT). When the OD$_{600}$ reached about 2.5, overexpression was induced by addition of 0.5 mM IPTG (final concentration) and cells were harvested after 3 h by centrifugation (10,000 × g, 10 min, 4 °C). For protein purification, cells were resuspended in 50 mM Tris-HCl pH 8.0, 300 mM NaCl, 1 mM phenylmethylsulfonyl fluoride, 17 µg/mL DNase I, 1% (v/v) Triton X-100, and 10 mM imidazole, and lysed by shear forces using a Microfluidizer (MP-110, Microfluidics) by passing them five rounds at 1500 bar. The solution was then centrifuged (30,000 × g, 30 min, 4 °C), and the supernatant was loaded onto Ni$^{2+}$-NTA resin (Qiagen) using a peristaltic pump. The resin was washed with each five column volumes of wash buffer-1 (40 mM Tris-HCl pH 8.0, 300 mM NaCl, 1% (v/v) Triton X-100), wash buffer-2 (40 mM Tris-HCl pH 8.0, 300 mM NaCl, 20 mM imidazole, 50 mM sodium cholate) and wash buffer-3 (40 mM Tris-HCl pH 8.0, 300 mM NaCl, 40 mM imidazole, 2.5 mM sodium cholate), and protein was eluted in wash buffer-3 with 300 mM imidazole. Fractions containing MSP were pooled, and the His-tag was cleaved by addition of Tobacco Etch Virus (TEV) protease (10:1 molar ratio, MSP:TEV) and dialyzed (10 kDa MWCO Spectra/Por dialysis tubing) against 20 mM Tris-HCl pH 8.0, 100 mM NaCl, 0.5 mM EDTA, 2.5 mM sodium cholate at 4 °C for 12 h. To remove His-tagged TEV protease and uncleaved MSP, Ni$^{2+}$-NTA resin (20% of the initial bed volume) was added and incubated for 90 min. The flow-through was collected and concentrated to ~5 mg/mL in an Amicon Ultra-15 10 kDa spin column concentrator (Millipore). Concentrated protein was aliquoted, snap-frozen in liquid nitrogen and stored at −80 °C until use.

### Reconstitution of IICB$^{Glc}$ into lipid nanodiscs

About 25 mg 1-palmitoyl-2-oleoyl-*sn*-glycero-3-phosphocholine (POPC) lipid (Avanti) dissolved in chloroform at 10 mg/mL was transferred to a round-bottomed flask, and the solvent was evaporated under a stream of nitrogen while rotating. The lipid film was further dried overnight in an exicator prior emulsion in 40 mM HEPES-NaOH pH 8.0, 150 mM NaCl. Emulsified lipids were then solubilized with 3% (w/v) DDM and diluted to a final lipid concentration of 10 mg/mL. To assemble IICB$^{Glc}$ into nanodiscs, freshly purified protein, untagged MSP1E3D1, and the POPC solution were diluted to a total volume of 300 µL of buffer (20 mM HEPES-NaOH pH 8.0, 150 mM NaCl, 5 mM β-ME, 5 mM glucose) and incubated on ice for 1 h, resulting in a final molar ratio of 1:5:100 (IICB$^{Glc}$:MSP1E3D1:POPC). Detergents were removed by the addition of 100 mg (wet weight) Bio-Beads SM-2 (Bio-Rad). The suspension was incubated on a rotating wheel at 4 °C for 1 h. Then, 100 mg of Bio-Beads were added and incubated overnight under gentle rotation. Nanodiscs were separated from the Bio-Beads using a syringe with a G-25 needle and the preparation was immediately incubated with pre-equilibrated Ni$^{2+}$-NTA resin (Qiagen, 200 µL bed volume) for 1 h at 4 °C. Empty nanodiscs and excess components were removed by washing the resin

in a gravity flow column (Promega) with 10 column volumes of wash buffer (20 mM HEPES-NaOH pH 8.0, 100 mM NaCl, 20 mM imidazole, 5 mM glucose), and the nanodisc-embedded IICB$^{Glc}$ was eluted with wash buffer containing 300 mM imidazole. The eluate was concentrated in an Amicon Ultra-0.5 100 kDa spin column concentrator (Millipore) and further purified by size-exclusion chromatography using a Superdex 200 10/300 GL column (Cytiva) equilibrated with detergent-free buffer containing 20 mM HEPES-NaOH pH 8.0, 100 mM NaCl, 5 mM glucose. Peak fractions were combined and concentrated using an Amicon Ultra-0.5 100 kDa spin column concentrator. The sample was supplemented with 0.1% (v/v) glycerol and centrifuged (20,000 × g, 10 min, 4 °C) before preparation of EM grids. Integrity of the sample was verified by SDS-PAGE and negative-stain EM.

### Electron microscopy sample preparation and cryo-EM data acquisition

Samples for negative-stain EM were prepared by adsorbing nanodisc-embedded IICB$^{Glc}$ at ~50 μg/mL on glow-discharged parlodion carbon-coated copper grids, washing three times in double-distilled water and then staining with 0.75% (w/v) uranyl formate. The grids were imaged on a 200 kV Tecnai F20 transmission electron microscope (FEI) equipped with a Falcon 3 direct electron detector and FEG (field emission gun). Samples for cryo-EM were prepared using a Vitrobot Mark IV (Thermo Fisher Scientific). The chamber was set to 100% humidity and 4 °C. 3 μL nanodisc-embedded IICB$^{Glc}$ at ~1 mg/mL was applied to glow-discharged (PELCO easiGlowTM system) Quantifoil R2/1 200-mesh copper grids. After blotting for 3 s with a blot force of −8, samples were vitrified by plunging into liquid ethane cooled by liquid nitrogen. High-resolution data collection was performed at 300 kV on a Titan Krios G2 (EMBL, Heidelberg, Germany) equipped with a BioQuantum K3 Summit direct electron detector (Gatan) at a nominal magnification of 130,000×. Movies were recorded and dose fractionated in 50 subframes with a total exposure time of 1.232 s (amounting to a total dose of 60.4 e$^-$/Å$^2$) and a physical pixel size of 0.645 Å. Micrographs were acquired using SerialEM software with a defocus range from −0.8 to −1.8 μm.

### Cryo-EM image processing

Data processing was performed using cryoSPARC version 3.2.0[31] as detailed in Supplementary Fig. 1g. 12,348 movies were subjected to motion correction and contrast transfer function (CTF) estimation using the patch-based implementations. Micrographs with CTF resolution >5 Å were discarded, and a subset of 500 micrographs was used to generate a model for the template picker using the blob picker followed by 2D classification. Template picking resulted in about 4.5 million particles, which were extracted (binned to 2.58 Å/pixel) using a box size of 384 pixels. Several rounds of 2D classification were used to remove featureless class averages yielding about 1.3 million particles, which were subjected to multi-class ab-initio reconstruction, requesting three classes and a maximum resolution of 6 Å. At this stage, the two conformations were observed and refined separately. The 570,086 particles corresponding to the symmetric volume were classified by heterogeneous refinements after re-extraction (binned to 1.032 Å/pixel), resulting in a homogenous particle stack with 237,565 particles. The volume was further refined by iterative rounds of non-uniform, homogenous and local refinements (imposing C$_2$ symmetry). A final round of non-uniform refinement[32], using the iteratively generated volume resulted in a global resolution of 2.56 Å at Fourier shell correlation (FSC) of 0.143 (gold standard). The 450,198 particles corresponding to the asymmetric volume were treated similarly for classification and refinement, but no symmetry was imposed. The final map was obtained using the local-refinement algorithm from 277,174 particles and had a global resolution of 2.89 Å. Local resolution of the final maps was assessed using cryoSPARC. For model building and visualization of 3D Coulomb potential map densities, the non-

sharpened half-maps were subjected to DeepEMhancer postprocessing package using the *highRes* learning model[33].

### Model building and refinement

The predicted IICB$^{Glc}$ AlphaFold model (ID: AF-P69786-F1; Supplementary Fig. 2g) was used as an initial atomic template for model building[34]. The monomeric model was fitted into the cryo-EM density map of the IF conformation using ChimeraX[35]. Several iterative rounds of manual adjustment were performed in Coot (version 0.9.6)[36], followed by real-space refinement in Phenix (version 1.19.2)[37]. Finally, the refined model was docked into the dimer map to generate the dimeric model, and the model was manually adjusted and refined. This model was then used to fit the OF conformation cryo-EM density map, and the model was built in the same manner as described above. The statistics of finalized models were validated using the MolProbity server and are reported in Supplementary Table 1. Cryo-EM density maps for transmembrane α-helices and the glucose binding sites with fitted models are provided in Supplementary Fig. 8.

### Molecular dynamics (MD) simulations

MD simulations were conducted to investigate the dynamics of the transporter, and the glucose recognition and binding mechanism. The asymmetric structure (IF-OF) was inserted into a 140 Å × 140 Å lipid bilayer composed of a mixture of 1-palmitoyl-2-oleoyl-sn-glycero-3-phosphoethanolamine and 1-palmitoyl-2-oleoyl-sn-glycero-3-phosphoglycerol (POPE:POPG ratio 3:1) assembled with the Membrane Builder from the CHARMM-GUI[38–40]. The system was solvated with a 25 Å water layer on each side of the lipid bilayer and neutralized with Na$^+$ and Cl$^-$ ions at a concentration of 150 mM (Supplementary Table 2). To enhance the probability of observing glucose binding events within the OF protomer, 20 glucose molecules were randomly placed in the water box, corresponding to a concentration of about 30 mM. The system was parametrized using the CHARMM36 force field[41–44] and the TIP3P model[45] was used for the water molecules. Simulations were performed with OpenMM 7.7.0[46], employing CHARMM-GUI's minimization and equilibration protocols. Electrostatic interactions were calculated using the Particle Mesh Ewald (PME) method. Force switching was employed for van der Waals interactions, with a switching distance set at 10 Å and a cutoff distance at 12 Å. Langevin dynamics with a friction coefficient of 1 ps$^{-1}$ were applied to maintain a temperature of 300 K, and constant pressure was ensured using the Monte Carlo barostat with a pressure of 1.0 bar[47]. To sample different initial binding positions of glucose, we initially utilized hydrogen mass repartitioning[48] to enable a 4 fs timestep. Five replicates were carried out for a cumulative simulation time of 4.85 μs. Since it has been demonstrated that the 4 fs timestep is suboptimal for sampling full-binding events[49], we subsequently selected 15 pre-binding events (a glucose molecule within 3 Å of residues H211 and H212) and employed them as seeds for simulations with a 2 fs timestep. These simulations were run until the glucose left the proximity of the binding site and correspond to a cumulative simulation time of 3.13 μs. We analyzed the entire production MD trajectories using VMD 1.9.4[50] with in-house tcl and python scripts. The occupancy of the hydrogen bonds was determined using the VMD HBonds Plugin[50] using an angle of 30° and a distance of 3.5 Å. For the kernel density estimate representation, the angle between the HP1b helix in the simulation with respect to the HP1b helix in the OF protomer of the cryo-EM structure was determined.

### Structure analysis and figures preparation

Protein interfaces were assessed using the PDBePISA server[51]. The IIC$^{Glc}$ protein was positioned in the membrane using the OPM server[52]. Alignments for r.m.s.d. calculations and pairwise deviation were performed in ChimeraX[35]. The HOLE 2.2.005 program[27] and the MOLE 2.5[53] server were used to analyse and visualize the accessibility of ligand

binding sites of cryo-EM and MD simulated models. Cryo-EM density maps and atomic models were visualized in ChimeraX, and figures were prepared using Adobe Illustrator (Adobe Inc.).

## Reporting summary

Further information on research design is available in the Nature Portfolio Reporting Summary linked to this article.

## Data availability

The data that support this study are available from the corresponding authors upon request. The cryo-EM maps have been deposited in the Electron Microscopy Data Bank (EMDB) under accession codes EMD-18640 (IF-IF state) and EMD-18641 (IF-OF state). The atomic coordinates have been deposited in the Protein Data Bank (PDB) under accession codes 8QSR (IF-IF state) and 8QST (IF-OF state). Source data (underlying Fig. 3b and Supplementary Fig. 1a, b) are provided in this paper. Additional data related to molecular dynamics simulations are available at Zenodo.org [https://doi.org/10.5281/zenodo.12748557]. Source data are provided with this paper.

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

## Acknowledgements

Screening of cryo-EM samples was performed on equipment provided by the Microscopy Imaging Centre (MIC), University of Bern, Switzerland. We thank the Electron Microscopy Core Facility (EMCF) at the European Molecular Biology Laboratory (EMBL) for their support in cryo-EM data acquisition. This work was supported by the University of Bern, the UniBern Forschungsstiftung (grant N° 8/2023), and the Swiss National Science Foundation (SNSF; grant N° 184980).

## Author contributions

Conceptualization: D.F. Data curation: P.R. and I.F. Funding acquisition: D.F. and T.L. Formal analysis: P.R., I.F., and T.L. Investigation: P.R, I.F., Z.U., and T.L. Methodology: P.R, I.F., Z.U., and T.L. Project administration: D.F. Resources: D.F. and T.L. Supervision: D.F., J.-M.J., and T.L. Writing – original draft: P.R. and D.F. Writing – review & editing: all authors.

## Competing interests

The authors declare no competing interests.
