## [Peer Review File · Nature Communications]

Structure and mechanism of a phosphotransferase system glucose transporterREVIEWER COMMENTS

Reviewer #1 (Remarks to the Author):

In the manuscript “Structure and mechanism of a phosphotransferase system glucose transporter”, the authors have addressed most of our concerns.

I have one major concern, which the authors have newly introduced in this manuscript regarding the supplementary figure 3 on the low resolution EIBC map. The authors claim that the signal shown in the 2D or 3D corresponds to the flexible EIB. However, EIB is only 8.3 kDa and 10 kDa if the linker is also included. It seems impossible to me that the density for EIB can have a similar size of the EIC plus micelle. It is possible that the "cloud", which has similar size of EIC in micelle, may represent mis-aligned neighboring particles on the micrograph. I would recommend removing this part from the manuscript, which will make the manuscript clearer. Once this is done, I can recommend publication of this manuscript in Nat. Communications.

Reviewer #2 (Remarks to the Author):

The authors use a series of classical molecular dynamics (MD) simulations to study the molecular basis and dynamics of substrate recognition and binding of EICBGlc. In total five replicas were used, leading to a total simulation time of $5.33 + 3.13 = 8.46 \mu\text{s}$. With respect to the technical aspects, the CHARMM36 force field was used and the MD engine was OpenMM.

While all simulation settings are state of the art, it is unclear why no enhanced sampling simulations were used to study the binding/unbinding events as described in the work. Similar calculations that focus on such (un)binding events nowadays are often falling back on enhanced sampling techniques such as metadynamics or LiGaMD (ligand gaussian-accelerated MD). The advantage of these approaches is that more robust statistical parameters can be derived for the binding events.

Second, the authors do not specify how the force field parameters of the glucose molecule were derived and how the partial charges of the glucose atoms were calculated (and what these are). Are these the one from the CHARMM36 FF, or where other parameters used? On a related point, the question remains in how far the results are dependent on the nature of the force field. It might be good practice to include a number of validation runs in which an alternative force fields, such as Amber94, is used to compare both results.

Third, different glucose analogues were used to examine the role of the different residues in substrate binding (Figure 3). Can these results be reproduced computationally? A potential manner to validate this would be to use relative binding free energy calculations in which one glucose analogue is “mutated” into another one, and calculating the associated change in binding free energy.

Remark: line 197: "Fig. 2b" should read "Fig 3b"?

Point-by-point response for Roth et al. (NCOMMS-24-16036A)

Reviewer #1

In the manuscript "Structure and mechanism of a phosphotransferase system glucose transporter", the authors have addressed most of our concerns.

Authors: We thank this Reviewer for appreciating our revisions.

I have one major concern, which the authors have newly introduced in this manuscript regarding the supplementary figure 3 on the low resolution EIIBC map. The authors claim that the signal shown in the 2D or 3D corresponds to the flexible EIIB. However, EIIB is only 8.3 kDa and 10 kDa if the linker is also included. It seems impossible to me that the density for EIIB can have a similar size of the EIIC plus micelle. It is possible that the "cloud", which has similar size of EIIC in micelle, may represent mis-aligned neighboring particles on the micrograph. I would recommend removing this part from the manuscript, which will make the manuscript clearer. Once this is done, I can recommend publication of this manuscript in Nat. Communications.

Authors: As requested by the Reviewer, we have removed Supplementary Figure 3 from the manuscript. The corresponding text passages in the manuscript have been removed or modified accordingly.

Reviewer #2

The authors use a series of classical molecular dynamics (MD) simulations to study the molecular basis and dynamics of substrate recognition and binding of EIICBGlc. In total five replicas were used, leading to a total simulation time of $5.33 + 3.13 = 8.46 \mu\text{s}$. With respect to the technical aspects, the CHARMM36 force field was used and the MD engine was OpenMM.

While all simulation settings are state of the art, it is unclear why no enhanced sampling simulations were used to study the binding/unbinding events as described in the work. Similar calculations that focus on such (un)binding events nowadays are often falling back on enhanced sampling techniques such as metadynamics or LiGaMD (ligand gaussian-accelerated MD). The advantage of these approaches is that more robust statistical parameters can be derived for the binding events.

Authors: This study focused on assessing the characteristics and stability of bound glucose within the EIIC-Glc complex. While enhanced sampling methods like metadynamics or LiGaMD could provide a more complete picture of the binding/unbinding energy landscape, these simulations demand significant computational resources and would go beyond the scope of this present study. Thus, the observed spontaneous release of glucose, even without enhanced sampling techniques, already offers valuable insights into the conformational flexibility at the binding pocket. In the revised version of our manuscript, we have refined and moderated our statements.

Second, the authors do not specify how the force field parameters of the glucose molecule were derived and how the partial charges of the glucose atoms were calculated (and what these are). Are these the one from the CHARMM36 FF, or where other parameters used?

Authors: We employed the standard parameters from the CHARMM36 force field (reference added to the manuscript) for the glucose molecule.

On a related point, the question remains in how far the results are dependent on the nature of the force field. It might be good practice to include a number of validation runs in which an alternative force fields, such as Amber94, is used to compare both results.

Authors: We opted for CHARMM36 due to its established strengths in modeling membrane bilayer structures, a crucial aspect. However, to partially address this concern, we performed brief simulations using Amber. As expected for a system with charged lipids, significant membrane distortions were observed (see Figure below).

E11C in POPE:POPG membrane bilayer simulated with the Amber force field (LIPID21) after 100 ns.

Third, different glucose analogues were used to examine the role of the different residues in substrate binding (Figure 3). Can these results be reproduced computationally? A potential manner to validate this would be to use relative binding free energy calculations in which one glucose analogue is “mutated” into another one, and calculating the associated change in binding free energy.

Authors: The observed hydrogen bond frequencies from the MD simulations (Figures 3c and d) demonstrate strong concordance with the experimental data on the different glucose analogues. This suggests that these simulations effectively capture the biochemically determined key interactions that stabilize glucose within the binding pocket.

Remark: line 197: “Fig. 2b” should read “Fig 3b”?

Authors: Thank you for pointing this out, this issue was corrected in the revised manuscript.

REVIEWERS' COMMENTS

Reviewer #2 (Remarks to the Author):

I believe that all remarks made by the reviewer are answered, and I can agree with publication of the manuscript.

Point-by-point response for Roth et al. (NCOMMS-24-16036A)

Reviewer #2

I believe that all remarks made by the reviewer are answered, and I can agree with publication of the manuscript.

Authors: We thank this Reviewer for appreciating our revisions.